Combined effect of Millet-Cowpea intercropping and biopesticide application against Heliocheilus albipunctella De Joannis (Lepidoptera: Noctuidae) in Burkina Faso

Boly Aboubacar bolyaboubacar@gmail.com 1 2
Waongo Antoine 1
Kabore Adama 3
Drabo Edouard 1
Traore Fousseni 1
Sanon Antoine 2
1 Laboratoire Central d’Entomologie Agricole de Kamboinsé, Institut de l’Environnement et de Recherches Agricoles, CREAF de Kamboinsé , Ouagadougou , Burkina Faso
2 Laboratoire d’Entomologie Fondamentale et Appliquée, Unité de Formation et de Recherches en Sciences de la Vie et de la Terre, Université Joseph Ki-Zerbo , Ouagadougou , Burkina Faso
3 Centre Universitaire de Dori, Université Thomas Sankara , Dori , Seno , Burkina Faso
Wu Huiting
Electronic publication date: 2025 Dec 2
Publication date: 2025
Volume: 13
Electronic Location ID: e20221
Received 2025 Mar 20; Accepted 2025 Sep 21
Copyright: ©2025 Boly et al.
Copyright year: 2025
Copyright holder: Boly et al.
License: This is an open access article distributed under the terms of the Creative Commons Attribution License, which permits unrestricted use, distribution, reproduction and adaptation in any medium and for any purpose provided that it is properly attributed. For attribution, the original author(s), title, publication source (PeerJ) and either DOI or URL of the article must be cited.
License URL: https://creativecommons.org/licenses/by/4.0/

Keywords: Pearl millet, Cultural control, Aqueous neem seed extract, Heliocheilus albipunctella, Burkina Faso

Funding: The McKnight Foundation, Global Collaboration for Resilient Food Systems of the McKnight Foundation, Minneapolis, MN 22-119; 2022–2025 The McKnight Foundation This work was supported by the McKnight Foundation, Global Collaboration for Resilient Food Systems of the McKnight Foundation, Minneapolis, MN (grant number 22-119; 2022–2025). The opinions expressed herein are those of the authors and do not necessarily reflect the views of the McKnight Foundation. The funders had no role in study design, data collection and analysis, decision to publish, or preparation of the manuscript.

==============================
Pearl millet, Pennisetum glaucum L. R. Br. (Poales: Poaceae), the main cereal crop in the Sahelian zone of Burkina Faso, is attacked by several insect pests, among which is the millet head miner, Heliocheilus albipunctella De Joannis (Lepidoptera: Noctuidae). Damage and yield losses caused by H. albipunctella on millet range from 30.00% to 85.00%. Control and management of H. albipunctella currently rely on synthetic insecticides, which are harmful to human and environmental health. Hence, there is a need to explore and develop alternative management strategies. Consequently, the current research, which was conducted, explored the use of millet-cowpea intercropping, a very common practice in the Sahelian zone of Burkina Faso, together with the application of biopesticides of Neem (Azadirachta indica A. Juss. (Sapindales: Meliaceae) seed kernels aqueous extracts. Fieldwork was carried out in Burkina Faso’s Djibasso and Dori communes during the 2021 rainy period. The obtained results found that the application of Neem extracts on cowpea plants at the flowering stage, synchronized with the heading stage of millet, significantly reduced the incidence of H. albipunctella. When millet was intercropped with cowpea, the application of aqueous extracts of Neem indirectly led to a significant reduction of about 50.00% in the number of larvae per spike. Additionally, a reduction in the percentage of millet spikes attacked, a decrease in mine length, and a gain in grain yield of more than 40.00% were observed. Thus, the findings from the application of this agricultural practice could be a promising control option against H. albipunctella.

Introduction

Pearl millet, Pennisetum glaucum L. R. Br. (Poales: Poaceae), is an ancestral cereal of critical importance in agriculture and nutrition for the Sahelian populations of West Africa (Gahukar & Ba, 2019; Dupuy, 2017). Under arid and semi-arid climatic conditions, millet is a vital resource for many rural communities (Shelke & Chavan, 2010). In Burkina Faso, pearl millet is the third most produced cereal after maize and sorghum, with an estimated production of 907,745.00 tons (Direction Générales des Etudes et des Statistiques Sectorielles (DGESS), 2023). Its high-protein content, energy value, vitamin and mineral composition are higher than those from other cereals such as wheat and maize (Parthasarathy Rao et al., 2006).

In the Sahelian region of Burkina Faso, millet is the mostly widely cultivated crop covering almost 80.00% of areas under cultivation due to its resistance to extreme climatic conditions and the dietary habits of the population (Saïdou, 2011; Gahukar & Ba, 2019). Heliocheilus albipunctella De Joannis (Lepidoptera: Noctuidae) is a pest that causes enormous damage to millet spikes in many sub-Saharan African countries, particularly Burkina Faso (Ndoye, 1991; Amadou et al., 2017). Damage is observed every year and is caused by larvae with grain yield losses of between 30.00 and 85.00% (Kaboré et al., 2017; Gahukar & Ba, 2019; Oumarou et al., 2019). Depending on the agroecological zones, millet is generally grown in association with several legumes, in particular cowpea, Vigna unguiculata L.Walp. (Fabales: Fabaceae) (Boly et al., 2022). This type of intercropping system is practiced by farmers with the aim of controlling diseases, weeds, and pests (Lawane et al., 2010; Guo et al., 2020). Likewise, it is used to increase the yield of cereals (Zoundi et al., 2007; Trail et al., 2016; Namatsheve et al., 2020). This intercropping is sometimes used in combination with other phytosanitary treatments of cowpea including the use of synthetic chemicals to control insect pests. However, the use of pesticides comes with the risks it poses to human, ecosystems, the environment, and loss of biodiversity (Carpentier, 2010; Barzman et al., 2015). Beyond the economic and environmental consequences, the massive and prolonged use of synthetic insecticides has also led to the development of resistance in several pests (Martin et al., 2000; Siddiqui et al., 2023). Considering all these reasons, research in Burkina Faso like in most part of the world, has over the last decade being geared towards the search for more ecofriendly strategies for managing insect pests, namely; biological control through the use of parasitoids (Ba et al., 2014; Kaboré et al., 2017), bio-pesticides and cultural practices. Thus, biopesticides associated with cultural practices could constitute an effective and promising alternative in the management of insect pests. Among the common biopesticides available, extracts from neem, Azadirachta indica A. Juss, containing Azadirachtin (Shafiq, Nadeem & Fazil, 2012; Kpindou et al., 2013) as its active ingredient has emerged as one of the most commonly used repellent due to its antifeedant effects on insects (Ngom et al., 2018; Bonni, Azonkpin & Paraïso, 2018). In addition to its repellent and insecticidal effect, it also has little or no effect on non-target species and has a low impact on the environment and biodiversity (Haseeb, Liu & Jones, 2004; Sanon et al., 2005). Indeed, it is used to control and manage more than 400.00 insect pest species including associated with crops (Tanzubil, Zakariah & Alem, 2004; Malick et al., 2008; Younoussou et al., 2021). However, it has not been used indirectly or in combination with other management strategies for cowpea or against H. albipunctella. Hence the reasons for this study were to evaluate the effectiveness of cultural control specifically intercropping in association with the biopesticide neem extract for the control and management of H. albipunctella in Burkina-Faso, With the specific aims of evaluating the indirect effect of cowpea treatment on the number of larvae per spike, H. albipunctella incidence and the grain yield of millet.

Material and methods

Study location

The study was conducted in Burkina Faso, in the communes of Djibasso and Dori during the 2021 rainy season. These two communes, Djibasso and Dori, are located in the Kossi and Seno provinces, respectively. Pearl millet is the main cereal crop in these two provinces and covers almost 78.00 and 80.00% of the cultivated area, respectively (Direction Générales des Etudes et des Statistiques Sectorielles (DGESS), 2023), and it is often intercropped with cowpea (Boly et al., 2022). The experimental plots were established in the villages of Bouakuy, located about 10 km from Djibasso, and Hoggo Sambowel, located about 10.00 km from Dori (Fig. 1). The cumulative rainfall from May to December was 827.00 mm in Djibasso and 557.00 mm in Dori. The relative humidity fluctuated between 53.16–91.44% in Djibasso and between 44.25–83.00% in Dori. While average monthly temperatures fluctuated between 25.00 and 32.00 °C. The vegetation is mostly covered with annual grass species, with areas of woodland and shrubland in which the dominant trees are Acacia species (Lykke, Kristensen & Ganaba, 2004). The soil in both communes is sandy in nature (Fontès & Guinko, 1995).

Figure 1 Map showing the villages selected for the implementation of the experimental plots in the communes of Djibasso and Dori.

Experimental design

The experimental design is composed of a divided plot (= Split Plot) made up of three sub-blocks (= Large plots). Each sub-block is composed of twelve (12.00) sub-plots (= Small plot). The sub-plots were each 9.60 m × 9.60 m in dimension (Fig. 2). This type of design provides the possibility of evaluating two factors in the system; the type of cropping association (primary factor) and the phytosanitary treatment (secondary factor). The cropping system consists of four combinations (i) millet (MP), (ii) cowpea monocultures as single main crop (NP), (iii) an intercrop consisting of two rows of millet and a single row of cowpea (2M-1N) and (iv) farmers’ practices (PP, one planting of cowpea between four plantings of millet). Such associations represent common practice among farmers in these regions. The four cropping systems form the main plot (Σ MP + NP + 2M-1N + PP). The phytosanitary treatment assigned to each main plot (Σ MP + NP + 2M−1N + PP) at each large plot level consisted of three combinations: (i) an aqueous neem seed extract (50 g/L; 5% w/v), containing a range of bioactive compounds, was used. Key constituents include azadirachtin (tetranortriterpenoid limonoid, C3 5H4 4O1 6), meliantriol (triterpenoid limonoid, C3 2H5 0O8), and salannin (triterpenoid limonoid, C3 4H4 4O9), which are primarily responsible for the extract’s insecticidal and repellent effects (Norten, 1999; Djibril et al., 2015); (ii) a synthetic insecticide (Lambda-Cyhalothrin 15.00 g/l + Acetamiprid 20.00 g/l); and (iii) a no-treatment control. Each of the cropping systems was replicated three times in each of the sub-blocks. All plots within each treatment were subjected to natural infestation by H. albipunctella. The spacings between rows and pockets in each cropping system were 0.80 m × 0.80 m for millet and 0.80 m × 0.40 m for cowpea. A distance of two m and six m was allowed between two sets of cropping systems and between sub-blocks respectively in order to prevent effects of treatments. For the millet, Local Djibasso and Local Dori were sown in the communes of Djibasso and Dori, respectively. The cowpea variety Komcalle was sown 20 days later to align its flowering stage (45 days after sowing) with the heading stage of millet. These varieties are the most commonly preferred and used by farmers in these areas. It is at this stage that H. albipunctella females prefer to lay their eggs on the spikes (Kabore et al., 2023). The sub-plots were thinned to two (2.00) millet plants per pocket at the first weeding, three weeks after sowing. A microdose of fertilizer consisting of 5.00 g of NPK (14/23/14; 100 kg/ha) per pocket was applied to both millet and cowpea after weeding followed by an application of urea (46% N; 50 kg/ha) to 3.00 g/pocket of millet at the time of the millet bolting.

Figure 2 Split-plot experimental design.

ANSE, Aqueous Neem Seed Extract. The vertical plane represents each of the three phytosanitary treatments applied across all types of cropping associations, which include four modalities: millet (MP), cowpea monocultures as single main crop (NP), two rows of millet alternating with one row of cowpea (2M+1N), and farmers’ practices (PP), consisting of one cowpea hill sown between four millet hills.

To prepare the neem aqueous extract, the neem seeds were collected under the trees, stored, and dried in the shade for 4 to 5 weeks. Next, they were ground into a fine powder, and this powder was macerated in water for 24 h. Finally, the macerated material was filtered to obtain the extracts (Dabiré-Binso, Ba & Sanon, 2008).

For the neem seed extract treatment subplots, a formulation of 500.00 g per 10 L (w/v) of water per 400.00 m2 was applied three times weekly (corresponding to 115 g per 2.30 L of water per subplot; Dabiré-Binso, Ba & Sanon, 2008). For synthetic insecticide treatment plots, a mixture of lambda-cyhalothrin (15.00 g/L) and acetamiprid (20.00 g/L) was applied twice at a rate of 1 L/ha, with applications two weeks apart (approximately 10 mL per 3.5 L of water per subplot). The choice of two treatments for the synthetic pesticide and three for the aqueous neem seed extract was based on previous studies in Burkina Faso (Dabiré-Binso, Ba & Sanon, 2008).

Data collection

Data collection started at the doughy grain stage of millet from each sub-plot and sub-block until harvest (time in weeks). The number of H. albipunctella larvae per spike, the damage caused by H. albipunctella (number and length of mines per spike and number of spikes attacked) and the grain yield were recorded. To determine the number of larvae per spike, an area of one m2 was delimited in each sub-plot and the number of larvae is counted on each spike contained in each surface and repeated four times. while the number of attacked spikes (spike bearing at least one mine), the number and length of mines per spike and the grain yield, an area of 9.00 m2 was determined at each sub-plot. Thus, the number of attacked spikes, the number and length of mines per spike and the grain yield were determined. The length of each mine was measured using a measuring tape. At harvest, the millet spikes from each area were threshed and weighed. The percentage of attacked spikes (PAE) and the grain yield (GY) were calculated for each sub-plot of each sub-block using Eqs. (1) and (2) (Boly, 2024): (1) PAE%=NSA/TNS×100.00;

where NSA: Number of spikes attacked per delimited area and TNS: Total number of spikes.

(2) GYkgha=GWS×10,000.00;

where GW: Grain weight of millet in kg per delimited area and S: delimited area in m2.

Data analyses

The data collected were analyzed using a two-way analysis of variance (Factorial ANOVA) to examine the influence of the phytosanitary treatments and the cropping systems (two independent variables) on the studied parameters. Post hoc tests were performed using the ’lsmeans’ package in RStudio (Lenth, 2016) when factorial ANOVA was significant between groups. The visualize the data, box plots and figure were plotted using R and Excel, respectively. All statistical analyses were carried out using R (R Core Team, 2022) and level of significance set at 5% for all statistical analyses.

Results

Influence of cropping system and phytosanitary treatment on the parameters studied

We found that phytosanitary treatment has a significant influence on the number of larvae per spike, H. albipunctella damage and grain yield (Table 1). The average percentage of damaged and the average number of damaged per spike were influenced by the cropping system (Table 1). On the other hand, the interaction between cropping system and phytosanitary treatments did not affect these parameters (Table 1).

Number of H. albipunctella larvae per spike according to phytosanitary treatments

The average number of larvae per spike of millet varied significantly depending on the location of the treatment (Djibasso ANOVA, F2.439 = 2.94; p = 0.003; and Dori ANOVA, F2.416 = 2.39; p = 0.01; Fig. 3). Regardless of the commune, it was greater in the sub-block that received no treatment (Fig. 3). In Djibasso, the average number of larvae per spike obtained with neem aqueous extract (1.33 larvae/spike) was significantly lower compared to treatment with Synthetic insecticide (where to find the result ie which figure and then the statistics, see above (ANOVA, F2.439 = 2.94; p = 0.003). In contrast, no significant difference was observed between the neem aqueous extract treatment and Synthetic insecticide in the commune of Dori (Fig. 3).

Table 1 Two-way analysis of variance was performed on the parameters studied in Djibasso and Dori to determine the effects of blocks, cropping association, phytosanitary treatment, and their interaction on the evaluated parameters.

Communes	Source of variation	Number of larvae per spike	Percentage of millet spike attacked	Number of mines per spike	Length of mines per spike	Grain yield	
Djibasso	Sub-blocks	–	–	0.89	0.94	–	
Cropping system	0.61	0.01	0.005	0.24	0.07	
Phytosanitary treatment	0.03	<0.0001	<0.0001	<0.0001	0.03	
Cropping system*phytosanitary treatment	0.52	0.13	0.64	0.21	0.19	
Dori	Sub-blocks	–	–	0.39	0.84	–	
Cropping system	0.65	0.09	0.09	0.44	0.86	
Phytosanitary treatment	0.03	<0.0001	0.19	0.005	0.004	
Cropping system*phytosanitary treatment	0.57	0.13	0.30	0.78	0.32	

Figure 3 Average number of larvae per spike, representing the infestation level of the larval population on the spikes in the communes of Djibasso and Dori according to the treatments different alphabetical letters on each point ind.

ANSE, Aqueous Neem Seed Extract.

Damage by H. albipunctella according to phytosanitary treatments

The average percentage of millet spike attacked by H. albipunctella was significantly greater at the level of the control treatment compared to those of the aqueous extract of neem and synthetic insecticide in the communes of Djibasso and Dori (Table 2). In addition, treatments with aqueous neem extract and synthetic insecticide reduced the average percentage of millet spike attacks by about 50.00% in both study sites. Regarding the average number of mines per spike, it varied significantly with location (Table 2). In Djibasso and Dori, the average number of mines per spike was lower from neem aqueous extract treatments in comparison to those from Synthetic insecticide and the control (Table 2). The mean length of mines, which reflects the extent of H. albipunctella damage, varied significantly at the level both communes (Table 2). In addition, in Djibasso and Dori, the mean length of mines per spike was lower at the level in the neem aqueous extract treatment than those of the synthetic insecticide and the control (Table 2).

Table 2 Percentage of millet spikes attacked (determined from the ratio of spikes with at least one mine to healthy spikes) and number of mines per spike in the communes of Djibasso and Dori according to the treatments.

Treatments	Percentage of millet spike attacked (% ± SE)	Number of mines per spike (Means ± SE)	Length of mines per spike (cm) (Means ± SE)	
	Commune of Djibasso	Commune of Dori	Commune of Djibasso	Commune of Dori	Commune of Djibasso	Commune of Dori	
Aqueous neem seed extract	37.23 ± 3.59b	1.59 ± 0.34b	2.30 ± 0.10c	1.15 ± 0.10a	12.81 ± 0.76b	3.19 ± 0.42b	
Synthetic insecticide	31.80 ± 3.14b	3.13 ± 0.78b	2.67 ± 0.12b	1.29 ± 0.09a	13.98 ± 0.89b	3.67 ± 0.33b	
Control	63.63 ± 7.04a	7.26 ± 0.70a	3.47 ± 0.14a	1.65 ± 0.14a	22.59 ± 1.31a	5.73 ± 0.53a	
Probabilities	F2,26 = 6.26p = 0.0006	F2,26 =  6.95p = 0.0003	F2,404 = 8.21p < 0.0001	F2,90 = 1.28p = 0.2666	F2,404 = 8.06p < 0.0001	F2,90 = 2.33p = 0.0394	
Notes.

Different alphabetical letters in the same column indicate a significant difference according to the pairwise mean comparison using the Tukey test (α = 0.05).

SE Standard Error

Effect of the cropping system on the percentage millet per spike attacked and the number of mines per spike in the commune of Djibasso

The percentage of millet spikes attacked and the number of mines per millet spike were significantly influenced by the cropping association system in the commune of Djibasso (Table 3).

The highest and lowest significantly different percentages of damaged millet spike were observed on millet grown alone and on the combination of two rows of millet and one row of cowpea (Table 3).

The average number of mines per spike was significantly higher in the farmer’s practice compared with millet grown alone and the combination of two rows of millet and one row of cowpea (Table 3).

Millet grain yield by phytosanitary treatments

Grain yield of millet varied significantly at the level of commune of Djibasso (ANOVA, F2.24 = 3.53; P = 0.0096; and Dori: F2.24 = 2.34; P = 0.00473; Fig. 4). In contrast, regardless of the commune, millet grain yield was statistically similar between the neem aqueous extract and Synthetic insecticide treatments (Fig. 4). The phytosanitary treatment, in both communes, made it possible to obtain more than one ton of millet per hectare compared to the control treatment which was less than one ton (Fig. 4).

Table 3 Percentage of millet spikes attacked (determined from the ratio of spikes with at least one mine to healthy spikes), and the number of larvae per millet spike, representing the infestation level of the larval population on the spikes according to the type of cropping association in the commune of Djibasso.

Cropping system	Percentage of millet spike attacked (% ± SE)	Number of mines per spike (Means ± SE)	
Farmers’ practice	42.63 ± 5.68ab	2.73 ± 0.17a	
2M-1N	35.93 ± 5.65b	2.11 ± 0.13b	
MP	54.10 ± 7.70a	2.04 ± 0.14b	
Probabilities	F2,24= 6.16 ; p = 0.01	F2,132= 6.16 ; p = 0.002	
Notes.

Different alphabetical letters in the same column indicate a significant difference according to the pairwise mean comparison using the Tukey test (α = 0.05).

SE Standard Error

Discussion

Millet cultivation, whether in monoculture or in association with cowpea, is a recurrent practice in Burkina Faso (Zoundi et al., 2007; Boly et al., 2022). This combination is sometimes accompanied by treatment of the cowpea with a synthetic insecticide or biopesticides to control insect pests of the cowpea crop. In contrast, the indirect effect of cowpea treatment on cereal insect pests associated with cowpea is sometimes overlooked.

In this study, we evaluated the indirect effect of cowpea treatment when intercropped with millet on the main pest of millet, H. albipunctella, in the field conditions. Our results show that the application of the aqueous extract of neem seeds on cowpea grown in association with millet at the time of heading gives similar results to those treated with the synthetic pesticide (Lambda-Cyhalothrin 15 g/l + Acetamiprid 20 g/l). These findings show a significant reduction in the number of H. albipunctella larvae per millet spike, the percentage of spikes attacked, and the length of mines in treated plots compared to the control. Synchronization cowpea treatment with the millet heading stage significantly reduced the activity of H. albipunctella females on treated sub-plots. This reduction in female activity led to lower infestations of millet spikes. Indeed, infestation of millet by H. albipunctella is conditioned by a synchronization between the period of heavy outbreaks of H. albipunctella and the sensitive stage of millet corresponding to the beginning heading stage (Bal, 1988). Likewise, H. albipunctella females prefer to lay their eggs at the heading stage, precisely at the top of millet spikes, with hatching occurring 3 to 4 days after oviposition (Ndoye, 1991; Kabore et al., 2023). The reduced infestation observed in the treated sub-plots may be attributed to the bioactive compounds in neem extract, such as azadirachtin, meliantriol, and salannin, which possess repellent, insecticidal, and fertility-reducing properties (Ascher, 1993; Abudulai et al., 2013). These compounds, with their repellent properties, can disrupt insect communication and disorient pest insects. They also regulate insect growth by affecting egg-laying behavior (Strickman, Frances & Debboun, 2009; Guèye et al., 2011). The efficacy of neem extracts against insect pests, particularly Lepidoptera, has been widely documented (Cherry & Nuessly, 2010; Bonni, Azonkpin & Paraïso, 2018; Ngom et al., 2018; Yao et al., 2022). Similarly, the repellent properties of neem extract against insect pests have also been demonstrated in beekeeping. Spraying neem leaf extracts around hives within a radius of five (5) meters, both before and after insect colonization, resulted in a significant reduction in the number of insects colonizing the hives (Gbedomon et al., 2012). Indeed, the repellent property observed with the treatment of neem extract could be attributed to the volatile compounds contained in A. indica (Belder Den, Elderson & Agca, 1998). These volatile compounds, present in neem seed extract, can deter females from locating their egg-laying sites, especially as their effect can last in the fields for 4 to 8 days depending on environmental conditions and the plant species treated (Schmutterer, 1990). The volatile compounds of neem are believed to act as an inhibitor, blocking the stimuli emitted by millet spikes, thereby deterring female H. albipunctella. Similar results were observed in Mali by Passerini (1991). The author reported a significant reduction in the number of eggs and mines of H. albipunctella after treating millet fields with an aqueous neem powder extract (Passerini, 1991). Moreover, volatile compounds are used in locating oviposition sites by females of H. albipunctella (Hall et al., 2001; Green, Owusu & Youm, 2004). The females prefer to lay eggs on spikes that have only emerged at 30.00% (Ndoye, 1991) which release high amounts of volatile compounds such as borneol that attract females. Thus, the mixture of volatile compounds from neem extracts with those emitted by the spikes in the same field of millet must have confused the females, who could no longer locate the spikes of millet for oviposition. This would explain the low number of larvae observed in the treated plots. Heavy rainfall can also reduce the density of H. albipunctella larvae, especially 1st and 2nd instars, which walk on the spikes. Indeed, the application of neem extract in the field resulted in a considerable reduction in the population density of Helicoverpa armigera Hübner (Lepidoptera: Noctuidae), Earias spp., Diparopsis watersi, Roths. (Lepidoptera :Noctuidae) Spodoptera littoralis (Lepidoptera :Noctuidae), and Syllepte derogata Fabricius (Lepidoptera : Crambidae) (Misra, Dash & Mahapatra, 2002; Abudulai et al., 2013; Douro et al., 2013).

Figure 4 Grain yield distribution of millet calculated for each treatment in the communes of Djibasso and Dori.

ANSE, Aqueous Neem Seed Extract. The boxplots with different alphabetical letters indicate significant differences according to the pairwise mean comparison using the Tukey test (α = 0.05).

We also found a significant reduction in the incidence of H. albipunctella, which were lower in the treated sub-plots than in the untreated ones. This reduction could also be explained by a low infestation of spikes observed in these sub-plots at the time of heading. Damage by H. albipunctella is mainly caused by larvae feeding on the floral organs, perforating the glumes, cutting the floral and fruit peduncles causing them to dry out. This result corroborates those of Ndoye (1991), who stipulates that the extent of damage caused by H. albipunctella on the millet crop would strongly depend on the coincidence between the flight of the adults and the period when earing begins in millet and the density of the larval population, making the effectiveness of phytosanitary treatment at the early heading stage against H. albipunctella very critical. This is why the application of endosulfan, Decis ULV (dimethoate + deltamethrin) and trichlorfon (dipterex + SI 8514) at early heading were effective against the H. albipunctella (Gahukar et al., 1986; Guèvremont, 1982). However, considering the harmful effects of these synthetic chemicals on humans, the environment, and biodiversity, it is advisable to use these products only if there is no alternative solution. Other groups of insect pests have been the subject of similar study. According to Adda et al. (2011), sowing Hyptis suaveolens (L) on the edges of maize fields halved the percentage of plants infested by Sesamia calamistis Hampson (Lepidoptera: Noctuidae).

In terms of the cropping system, the combination of two rows of millet and one row of cowpea resulted in a significant reduction in the percentage of millet spikes that were attacked and the average number of mines per spike. This reduction can be attributed to the diversity of the landscape, which not only disorients females in finding oviposition sites, but also provides a refuge for natural enemies. Indeed, pests may lose the ability to locate host plants when confronted with a mixture of several volatiles from non-host plants. This hypothesis supports the assertion that volatile compounds are important for the location of oviposition sites of the MEM (Hall et al., 2001; Green, Owusu & Youm, 2004).

The results of the present study also showed the effectiveness of neem extract in increasing grain yield of millet by more than 40.00%. The increase in grain yield observed in the treated plots could be explained by a low number of H. albipunctella larvae. Indeed, according to Gahukar & Ba (2019) and Oumarou et al. (2019), yield losses due to H. albipunctella larvae attacks without any external intervention in the Sahelian zone are estimated between 30.00 and 80.00%. The results of this study show that, in the context of integrated pest management, treating cowpea in association with millet using neem seed extract can protect the millet crop against H. albipunctella. Neem seed kernel extract could constitute a promising alternative to synthetic pesticides. This solution is all the more relevant as neem seeds are widely available in these regions and easily accessible to producers. They only need to collect mature seeds fallen at the foot of the trees, dry them in the shade, and then grind them to obtain a powder. This approach could also present the advantage of being less costly than purchasing synthetic products.

Conclusion

This study on the efficacy of the crop association associated with neem extract in the control of the millet head miner, H. albipunctella showed a particular interest in the importance of treating cowpea in association with millet in the field. Indeed, the use of aqueous neem seed extract on cowpea in combination with millet significantly reduced the infestation of millet by H. albipunctella. This resulted in a significant reduction in the percentage of millet ears attacked, the number and length of mines, and an increase in productivity. For these reasons, the use of this biopesticides in the treatment of cowpea in association with millet could constitute an alternative control method for the control of the millet head miner in the Sahelian zone.

Supplemental Information

Supplemental Information 1 Data for the Djibasso municipality including records on the number of larvae per ear, the number and length of mines, as well as millet grain yields

Supplemental Information 2 Data for the Dori municipality, containing measurements of the number of larvae per ear, mine number and length, and grain yield

Additional Information and Declarations

Competing Interests

Author Contributions

Data Availability

The authors declare there are no competing interests

Aboubacar Boly conceived and designed the experiments, performed the experiments, analyzed the data, prepared figures and/or tables, authored or reviewed drafts of the article, and approved the final draft.

Antoine Waongo conceived and designed the experiments, performed the experiments, analyzed the data, prepared figures and/or tables, authored or reviewed drafts of the article, and approved the final draft.

Adama Kabore conceived and designed the experiments, performed the experiments, analyzed the data, prepared figures and/or tables, authored or reviewed drafts of the article, and approved the final draft.

Edouard Drabo conceived and designed the experiments, performed the experiments, prepared figures and/or tables, authored or reviewed drafts of the article, and approved the final draft.

Fousseni Traore conceived and designed the experiments, authored or reviewed drafts of the article, and approved the final draft.

Antoine Sanon conceived and designed the experiments, authored or reviewed drafts of the article, and approved the final draft.

The following information was supplied regarding data availability:

The data can be found in the Supplemental Information.

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

Younoussou et al. (2021) Younoussou R Abdou MA Abdou L Mahamane A 2021 Effets comparés de quelques biopesticides et d’un pesticide chimique (cyperméthrine 10 EC) sur les insectes ravageurs et maladies parasitaires du Niébé, Vigna unguiculata (L.) Walp (Fabaceae) Revue Marocaine des Sciences Agronomiques et Vétérinaires 9 4 710 717
Zoundi et al. (2007) Zoundi JS Lalaba A Tiendrébéogo JP Bambara D 2007 Systèmes de cultures améliorés à base de niébé (Vigna unguiculata (L.) Walp) pour une meilleure gestion de la sécurité alimentaire et des ressources naturelles en zone semi-aride du Burkina Faso TROPICULTURA 25 2 87 96