# Peer review of "Combined effect of Millet-Cowpea intercropping and biopesticide application against Heliocheilus albipunctella De Joannis (Lepidoptera: Noctuidae) in Burkina Faso"

_PeerJ, doi:10.7717/peerj.20221_

## Round 0.1 · original submission · Major Revisions

· Academic Editor

Major Revisions

**Language Note:** The review process has identified that the English language must be improved. PeerJ can provide language editing services - please contact us at [email protected] for pricing (be sure to provide your manuscript number and title). Alternatively, you should make your own arrangements to improve the language quality and provide details in your response letter. – PeerJ Staff

·

Basic reporting

This paper presents an original research study. The research topic focuses on the biotic factors that affect Pearl millet, one of the main food crops in the Sahelian zone. The research involves the application of cultural practices as a tool for managing and controlling the millet head miner, Heliocheilus albipunctella De Joannis (Lepidoptera: Noctuidae).

Title, Abstract & Introduction:
The title, abstract, and introduction are well-suited to the research content.

Author.
1. Follow carefully the research methodology
2. Did a statistical analysis of the data

Results &Discussion:
Done, written, and displayed excellently.

Figure
If possible to be in colour for better demonstration.

Tables:
Reflect the results

References:
The references should be revised and rewritten because of:
1. Many missing authors in the reference section
Example: Gahukar & Ba, 2019; Dupuy, 2017)
2. Many of the citations, if written by many authors in the text, only one appears in the references section;
examples: Ndoye, 1991; Amadou et al., 2017; Gahukar and Ba, 2019

Experimental design

The choice of the experimental design is suitable & allows a study of several factors that can have on a response.

Validity of the findings

The research findings are valid and reliable.

·

Basic reporting

The study highlights significant yield losses attributed to this pest and analyzes the difficulties the millet head miner, H. albipunctella, poses to pearl millet crops in Burkina Faso. The study investigates a different strategy that uses neem-based biopesticides in conjunction with millet-cowpea intercropping. According to research, applying neem extract at particular stages of crop growth reduced pest incidence, larval populations, and related crop damage while also increasing grain yield. According to the authors, combining botanical extracts with conventional intercropping techniques could provide a workable management solution in this situation. Additionally, there are numerous formatting issues, typographical errors, outdated references, and inconsistencies with journal style throughout the manuscript that require attention.

Experimental design

• Lines 93–93: Please clarify the relevance of the listed plant species to the current study. Are these species hosts of the target insect pest, part of the experimental setup, or referenced for comparative purposes? Their inclusion requires context to establish their relationship with the study’s objectives.

• Lines 111–113: Please indicate the aqueous neem seed extract concentration used in the experiment. Furthermore, the reference used to describe its chemical makeup is fairly out of date. It would be more appropriate to cite a recent, region-specific study or to present a chemical analysis pertinent to the current experimental location, as the phytochemical profile of neem can vary greatly depending on geographic and environmental factors.

• Lines 119–121: Kindly give the exact variety of the host plant that was used in the experiments. This information is crucial for reproducibility as well as for realizing the interplay between plant, pest, and treatment.

• Line 127: The preparation method of neem extract is incomplete, with vital information missing. Please provide the precise amount of neem seeds and the amount of water utilized in the preparation of the extract so that the protocol can be easily replicated.

• Line 122: The manuscript does not provide any information about the source and upkeep of the insect pest employed in the experiments. Please explain if the insects were taken from the field or reared in a controlled environment. If taken from the field, mention what precautions were taken to avoid pre-experimental oviposition on host plants.

• Lines 135–147: Although the protocol for data collection is defined, its scientific applicability and acceptability need referencing and backing. Provide citations for formulas and methodologies applied in calculating parameters like mortality, oviposition, or deterrence indices, to authenticate the method employed.

• Line 141: The line seems to have a typing error and must be deleted or corrected for clarity.

Validity of the findings

• Results Section: The authors have not provided the actual numerical values or summary statistics of the experimental findings in the results section. Including mean values, standard deviations/errors, and statistical significance (e.g., p-values) is essential for interpreting the outcomes and assessing the study's scientific merit.

• Lack of Scientific Elaboration: The findings are not presented in a way that is thorough or consistent with science. Please provide more details about the results by going into detail about the trends, treatment effects, statistical results, and biological significance. Steer clear of ambiguous statements and make sure your interpretation of the data is well-supported.

• Study Justification and Originality: The study seems to be a straightforward inquiry with generally expected findings. Particularly in the Introduction, Results, and Discussion sections, the manuscript lacks a well-defined rationale. The authors need to do a better job of explaining the scientific significance of the findings, the novelty of their approach, and the research gap they addressed.

• Terminology Inconsistency with Positive Control: Although the positive control, a synthetic insecticide, is mentioned in the text, it is referred to as "K_optimal" in the accompanying table. This inconsistency may cause confusion. Please standardize the terminology throughout the manuscript to maintain clarity and coherence.

• Section 3.4: The content in Section 3.4 is presented as short snippets. For better readability and scientific presentation, kindly revise this section into a cohesive paragraph format.

• Line Formatting: In lines 202, 205, etc., the letter p used for significance values is not italicized. Please correct this according to standard scientific formatting conventions (e.g., p < 0.05).

Additional comments

• Line 48 – Taxonomic Details: Please provide the full scientific name, including the genus, species, and taxonomic family, when mentioning the organism for the first time in the manuscript. This will improve clarity and taxonomic accuracy.

• Lines 52–60 – Rephrasing for Clarity: These lines are difficult to follow due to awkward phrasing and lack of logical flow. Kindly rephrase them to improve readability and ensure a coherent connection between the statements.

• Line 78 – Typographical Error: This line contains a typographical error and should be removed or corrected.

• Lines 123 & 130 – Outdated References: The references cited here are quite old. Given the advancements in this area, please consider replacing them with more recent and relevant studies to reflect the current state of research.

• Line 208 – Typographical Error: There is a typographical error on this line that needs correction.

• Table I – Formatting of ‘NA’: Please use a dash (–) instead of “NA” to indicate data not available or not applicable, as per common scientific formatting conventions.

• Tables II & III – Abbreviation Correction: The abbreviation “ES – Standard Error” should be corrected to “SE – Standard Error.” This appears to be a typographical error.

• Outdated References – General Comment: Many of the cited references throughout the manuscript are outdated. Please update the reference list by including recent studies published in the last 5–10 years to strengthen the relevance and impact of the work.

• Reference Formatting – Inconsistencies: Several references do not follow a consistent citation style. Kindly revise the reference list according to the journal’s formatting guidelines to ensure uniformity and professionalism.

·

Basic reporting

-The article as a whole should be carefully reviewed in terms of grammar and phrasing.

-Especially, the figures should be prepared more professionally.

- Revise for conciseness and flow, especially in the Introduction and Discussion.

- Add a brief limitation statement regarding environmental variability (rainfall differences between Djibasso/Dori).

Experimental design

- Indicate how standardization between plots was achieved in experiments conducted in a relatively large area. In other words, how soil homogeneity was achieved.

- Justify why you chose the "Komcallé" variety (pest resistance characteristics, local preference, etc.).

- Provide a well-prepared diagram for a better understanding of the experimental plan.

- Write in more detail the reason for the 3-week interval between neem applications.

- Discuss how the intercropping rates (2M-1N) were determined (previous studies or local practices?).

Validity of the findings

The findings are statistically valid, and the conclusions are comprehensively stated. The study can be a good reference for future studies on indirect pest control via intercropping. No major revisions are needed for validity. Only a minor addition can be made to the following topic.

- Briefly discuss the economic feasibility (cost and yield benefits) of neem for farmers.

---

## Round 0.2 · accepted · Accept

· Academic Editor

Accept

The authors have addressed all of the reviewers' comments, and this manuscript is ready for publication.